# Identification and Characterization of a Phosphate-Solubilizing Bacterium and Its Growth-Promoting Effect on Moso Bamboo Seedlings

**Yang Zhang [1], Songze Wan [2,3] , Fuxi Shi [2,3], Xiangmin Fang [2] and Chao Huang [2,*]**

[1] Jiangxi Provincial Key Laboratory of Conservation Biology, Jiangxi Agricultural University, Nanchang 330045, China; zhangyang0558@jxau.edu.cn

[2] Key Laboratory of National Forestry and Grassland Administration for the Protection and Restoration of Forest Ecosystem in Poyang Lake Basin, Jiangxi Agricultural University, Nanchang 330045, China; swan0722@jxau.edu.cn (S.W.)

[3] Matoushan Observation and Research Station of Forest Ecosystem, Fuzhou 335300, China

[*] Correspondence: huangchao85@jxau.edu.cn

**Abstract:** Phosphate-solubilizing bacteria (PSB) offer an eco-friendly approach to boost plant growth in soils low or deficient in phosphorus (P). In this study, we isolated 97 PSB strains from the soil around moso bamboo roots in Jiangxi Province, China. The RW37 strain was identified as *Enterobacter soli* through its physical characteristics and genetic sequencing. Our experiments revealed that RW37 could dissolve phosphate at levels exceeding 400 mg $L^{-1}$ across a wide range of environmental conditions, including temperature (25–35 °C), pH levels (3.5–7.2), salinities (0–2.0%), and volumes of medium (1/5–3/5 of flask volume), showcasing its adaptability. Furthermore, RW37 showed remarkable phosphate-solubilizing abilities at various pH levels using different phosphate sources, with the highest capacity observed in a medium containing $CaHPO_4$. This study also found a negative correlation between P-solubilizing capacity and fermentation broth pH, indicating that RW37 likely secretes organic acids to dissolve phosphate sources. Pot experiments demonstrated that applying RW37 significantly improved the plant height, biomass, root growth, and P uptake of moso bamboo seedlings in red soil. Our results highlight the potential of RW37 as an eco-friendly biofertilizer for subtropical bamboo forests.

**Keywords:** *Enterobacter soli*; *Phyllostachys edulis*; P solubilization ability; growth-promoting effect; microbial fertilizer

## 1. Introduction

Phosphorus (P) is a vital nutrient essential for plant growth, development, and reproduction [1,2]. Despite its abundance in soil, challenges arise due to the low solubility of mineral P and its fixation within iron complexes, hindering efficient P uptake by plants. More than 95% of soil P in China is insoluble [3,4]. Chemical P fertilizers have been extensively utilized in agricultural and forestry ecosystems for nearly two decades to address this issue [5]. However, the excessive application of P fertilizers escalates production costs and adversely affects soil quality [6]. Consequently, employing environmentally friendly strategies to enhance P absorption and utilization in crops holds significant ecological and economic importance.

Previous studies have shown that root-associated microorganisms positively impact the utilization of plant soil nutrients [7]. Microorganisms can enhance the effectiveness of P in soil [5], so considering the influence of microorganisms on P cycling in soil is crucial. For example, phosphate-solubilizing microorganisms (PSM) promote plant growth by providing soluble P [8]. Among them, phosphate-solubilizing bacteria (PSB) have received widespread attention as an essential component of PSM [8]. Some inorganic

phosphate-solubilizing bacteria can effectively convert fixed P into a biologically available form and prevent the re-fixation of released P [9]. Previous studies have identified bacterial strains from the genera *Pseudomonas*, *Bacillus*, *Rhizobium*, and *Enterobacter* as potent P mobilizers in soil, positively affecting plant growth [10–12]. However, PSB strains isolated from different soils may exhibit functional differences in their P solubilization abilities [4]. Therefore, isolating and evaluating the environmental adaptability of PSB can provide valuable information for the practical application of PSB strains.

Moso bamboo (*Phyllostachys edulis*) is a common bamboo species in China, covering approximately 70% of the bamboo forest area in China and 80% globally [13,14], totaling around 3.87 million hectares. Moso bamboo has become an important economic tree, meeting industrial demands and increasing farmers' income [15,16]. However, the extensive management practices over a long period have led to a decline in soil fertility and insufficient soil nutrients, limiting the productivity of moso bamboo. To enhance the growth and yield of moso bamboo, extensive fertilizer applications have been employed, with P identified as the primary limiting factor for moso bamboo productivity [3,17]. Although fertilization promotes moso bamboo growth, the associated increase in costs and the negative environmental impact of chemical fertilizers have prompted researchers to seek alternative, eco-friendly technologies [11,17].

The purpose of this study is to isolate new strains capable of effectively solubilizing insoluble phosphates in the rhizosphere soil of moso bamboo and to attempt to evaluate their P-solubilizing activity and potential mechanisms, as well as their effects on the growth of moso bamboo seedlings. Our research findings are expected to enhance the understanding of the characterization and related mechanisms of PSB in solubilizing insoluble phosphate, and to provide promising clues for the potential application of PSB in P-deficient areas and acidic soils.

## 2. Materials and Methods

### 2.1. Sample Collection

Rhizosphere soils were sampled between July and September 2015 from Dagang Forest Farm (28°37′17″ N and 114°56′31″ E), Guanshan Forest Farm (28°33′35″ N and 114°34′47″ E), and Dajing Forest Farm (26°34′8″ N and 114°8′19″ E), in Jiangxi Province, China. The basic characteristics of the field site and the physicochemical properties of the rhizosphere soil are provided in Table 1.

**Table 1.** Basic parameters and soil characteristics of the three sampling sites.

| Sampling Sites | Elevation (m) | pH | Total N (g·kg$^{-1}$) | Total P (g·kg$^{-1}$) |
|---|---|---|---|---|
| Dagang | 381.0 | 5.16 ± 0.09 b | 4.65 ± 0.10 c | 0.27 ± 0.03 a |
| Guanshan | 548.0 | 4.37 ± 0.07 a | 2.51 ± 0.09 a | 0.33 ± 0.04 a |
| Dajing | 1105.0 | 4.39 ± 0.06 a | 3.44 ± 0.08 b | 0.54 ± 0.07 b |

Notes: Value = Mean ± standard error. N = 5. The different letters indicated the significant differences among the three sampled sites at the level of $p < 0.05$.

To collect rhizosphere soil at each sampling site, five representative moso bamboo plants were selected on the basis of the average diameter at the breast height (DBH) and height of trees (H) with 5- or 6-year-olds. The living roots of a moso bamboo were carefully removed and gently shaken by hand. The soil attached to the surface of the root of approximately 4 mm in size was regarded as rhizosphere soil. All rhizosphere soils from a moso bamboo were mixed into one soil sample, and all soil samples were immediately taken to the laboratory.

### 2.2. Isolation of PSB

Approximately 10 g of soil from each soil sample was transferred into an Erlenmeyer flask containing 90 mL of sterile water. The mixture was then shaken at a speed of 120 revolutions per min for a duration of 60 min to ensure proper mixing. Afterward, a series

of 10-fold diluted suspensions were prepared, and 200 μL of each dilution was plated on Pikovskaya's agar composed of 10 g glucose, 0.2 g NaCl, 0.2 g KCl, 0.5 g $(NH_4)_2SO_4$, 0.1 g $MgSO_47H_2O$, 0.5 g $FeSO_47H_2O$, 0.5 g $MnSO_47H_2O$, 0.5 g yeast extract, and 18 g agar in 1000 mL of distilled water, and 0.5% tricalcium phosphate $(Ca_3(PO_4)_2)$ was used as an insoluble P source [18]. The presence of a clear halo around colonies after 5 days of incubation at 30 °C was used to identify the PSB [19]. The experiments were conducted in triplicate, and PSB was isolated and purified by culturing on Luria broth (LB) agar medium (10 g/L tryptone, 5 g/L yeast extract, 10 g/L NaCl, 1 L deionized water, and pH = 7.2) at 30 °C based on the size of halo zones.

### 2.3. Identification of PSB

The identification of PSB isolates involved analyzing their physiological and biochemical characteristics, following the methods outlined in Bergey's Manual of Systematic Bacteriology [20]. Additionally, genotypic identification was carried out by amplifying and partially sequencing the 16S rRNA [21]. To amplify the 16S rRNA gene fragment, bacterial universal primers 27F (5′-GAGTTTGATCACTGGCTCAG-3′) and 1492R (5′-TACGGCTACCTTGTTACGACTT-3′) were used for polymerase chain reaction (PCR) [22]. The PCR reaction mixture (20 μL) contained 1.0 μL of genomic DNA (10 ng/μL), 8 μL of 2× PCR MasterMix buffer (0.05 μg/μL Taq polymerase, 4 mM $MgCl_2$, and 0.4 mM dNTPs), 0.5 μL of each primer (10 pmol/μL), and 10 μL of ultrapure sterile water. The amplification protocol included one cycle of 5 min at 94 °C, followed by 30 cycles of 30 s at 94 °C, 30 s at 56 °C, and 1 min at 72 °C, and then one cycle of 5 min at 72 °C [23]. The model of instrument was PCR T100 and was produced by Bio-Rad Laboratories, Inc. Following amplification, the PCR products underwent sequencing. The 16S rRNA gene sequence was compared to the GenBank database using BLAST (Basic Local Alignment Search Tool) at NCBI. The resulting alignment was used to construct a phylogenetic tree via the neighbor-joining method using MEGA 6.0 (Molecular Evolutionary Genetics Analysis) software [24].

### 2.4. Effects of Environmental Factors on the Phosphate-Solubilizing Activity of PSB

The phosphate solubility of isolated PSB under different environmental conditions was determined. Five temperatures (20 °C, 25 °C, 30 °C, 35 °C, and 40 °C) were controlled using an incubator shaker (Yi Heng, China). Nine salinities (0%, 1%, 2%, 3%, 4%, 5%, 6%, 7%, and 8%, *m/v*) were adjusted using pure NaCl to assess the salinity tolerance of RW37. Five volumes of medium (1/5, 2/5, 1/2, 3/5, and 4/5) were established in 100 mL Erlenmeyer flasks. Moreover, the P-solubilizing capacity of five P sources under different initial pHs of RW37 was also measured. Next, 100 mL flasks containing 50 mL of NBRIP were prepared. The P in the medium was substituted with the following five insoluble P sources at 200 mg P $L^{-1}$: $Ca_3(PO_4)_2$, $FePO_4$, $CaHPO_4$, $AlPO_4$, or $C_6H_6Ca_6O_{24}P_6$ (phytate). The pH of the five P treatments was adjusted to 2.5, 3.5, 4.5, 5.5, 6.5, and 7.2 with 2 mol/L hydrochloric acid after the addition of the P sources, and the medium was sterilized at 121 °C for 20 min. One milliliter of bacterial seed liquid was added to provide $10^8$ CFU/mL. After 96 h of incubation, the supernatants were collected to measure soluble P and pH via centrifugation, and the soluble P was determined using the molybdenum–antimony colorimetry method [25]. Each treatment was inoculated with bacteria in triplicate. Culture media without bacterial inoculation served as a control.

### 2.5. Characterization of Indole Acetic Acid (IAA) Production

The IAA production of the isolated PSB was determined using the Van Urk–Salkowski reagent method [26]. PSB was grown in LB medium supplemented with 0, 2, and 5 mg $mL^{-1}$ of tryptophan and incubated at 30 °C for 12 days. Results were obtained after 6 and 12 days of incubation. The fermentation broth was centrifuged at 10,000 rpm for 10 min. The model of device was eppendorf 5804 and was produced by Eppendorf China Ltd. (Shanghai, China). Then, 1 mL of the supernatant and 2 mL of Salkowski's solution, which includes 0.5 mol $L^{-1}$ $FeCl_3$ and 35% of $HClO_4$, were mixed and incubated in the dark at 40 °C for

30 min. Finally, IAA was measured using a colorimetric technique at 533 nm and calculated from the standard curve with known concentrations of pure IAA that were purchased.

### 2.6. Inoculation on Moso Bamboo Seedlings

The soil used for plant cultivation in the present study was collected from Meiling Mountain in Jiangxi Province, China (115°45 N, 28°47 E). The soil was sieved (2 mm) and air-dried before being mixed with coarse sand and vermiculite at a ratio of 2:1:1 (*W*:*W*:*W*). The mixed soil was sterilized and had a total N content of 0.46 g kg$^{-1}$, a total P content of 0.22 g kg$^{-1}$, an organic matter content of 6.43 g kg$^{-1}$, and a pH of 5.92. One-month-old moso bamboo seedlings taken from the sand table were then planted in plastic pots containing the sterilized soil and were inoculated with PSB or left as a control without inoculation. Inoculation with the isolate was performed by applying 10 mL of an aqueous suspension containing $1 \times 10^8$ CFU mL$^{-1}$ on the surface of each pot at various points around the seedlings [27]. The experiment was conducted with five replications in a completely randomized design. To maintain a consistent microbial inoculate load, each pot containing moso bamboo seedling plant was inoculated using the same methods after 60 days. The seedlings were grown in a glasshouse for 180 days, with a day/night temperature of 20/15 °C and appropriate watering. At 180 days after inoculation, the plant ground diameters (D) and shoot heights (H) were measured. The increase rates of shoot heights and ground diameters for both the inoculation treatment and control were calculated according to the following formulas: increase rate (%) = $(H_{180} - H_0)/H_0 \times 100\%$ for shoot heights, and increase rate (%) = $(D_{180} - D_0)/D_0 \times 100\%$ for ground diameters.

### 2.7. Chemical Analysis of Soil and Plant Samples

Moso bamboo was dissected into root, stem, and leaf samples and washed with distilled water. Fresh subsamples were then oven-dried at 65 °C for 72 h to determine the dry weight (DW) and P content in the plant's tissues. Total P was determined using the Kjeldahl method and the molybdenum–antimony colorimetry method after the samples were digested with $H_2SO_4$ [28]. Five random soil samples were collected from the rhizosphere of moso bamboo seedlings. Soil pH was measured using a PHS-3C pH meter (Shanghai Lida Instrument Factory, Shanghai, China), with a soil to water ratio of 1:2.5. Soil $NH_4^+$-N and $NO_3^-$-N were extracted with 2 M KCl for 30 min and then measured via spectrophotometry using the indophenol blue and cadmium reduction methods, respectively. Available soil P was extracted with 0.5 M $NaHCO_3$ for 30 min, and its concentration was determined using the molybdenum–antimony colorimetry method [1].

### 2.8. Statistical Analysis

Statistical analyses were conducted using SPSS package version 16.0. The data were presented as mean ± standard error (SE). Student's *t*-test analysis and one-way ANOVA were employed to calculate data variance, with $p \leq 0.05$ indicating a significant difference.

## 3. Results

### 3.1. Isolation and Identification of PSB

In this study, a total of 97 PSB isolates were collected from three sites containing fifteen soil samples. Among these, isolate RW37 was identified as a potent phosphate solubilizer, exhibiting the largest clear halo zone around its colony (Figure 1A). The biological characteristics of RW37 were recorded 48 h after inoculation, and the colonies were photographed (Figure 1B).

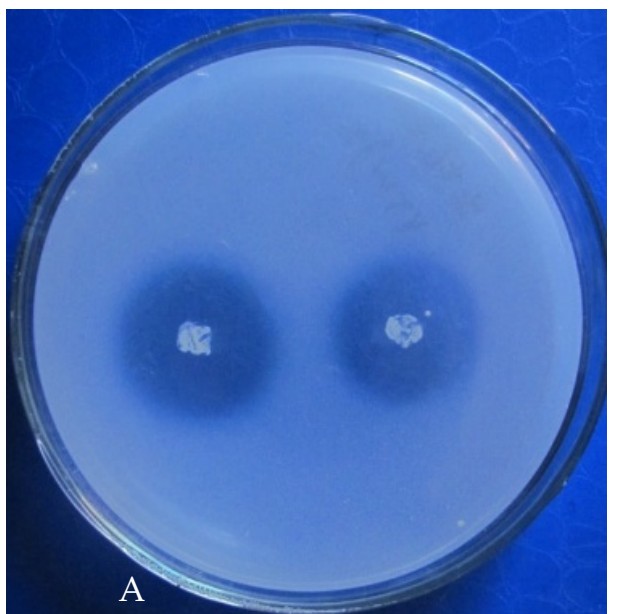
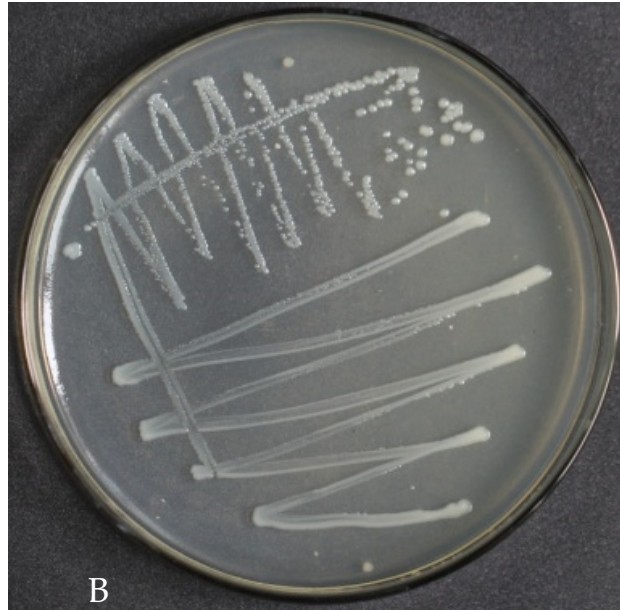

**Figure 1.** Zones of phosphate solubilization and biological characteristics of the RW37 strain ((**A**): Zones of phosphate solubilization on NBRIP agar plates; (**B**): Colony characteristics on LB plate).

The strain formed convex and transparent colonies with a circular shape and neat edges. The surface of colonies was smooth, glossy, and wet. The strain was identified as a Gram-negative bacterium with short rod-shaped cells using the Gram reaction and KOH test. The aerobic nature of the strain was confirmed, as no growth was observed under the coverslip during the respiration type test (Table 2).

**Table 2.** Biochemical characteristics of the isolates of the RW37 strain.

| Biochemical Reactions | RW37 |
| --- | --- |
| 3% KOH solubility | + |
| Contact enzyme | + |
| Oxidase | − |
| Gelatin hydrolysis | + |
| Starch hydrolysis | + |
| Citrate | + |
| Indole production | − |
| Methyl red | + |
| Nitrate reduction | + |
| Malonate utilize | + |
| V-P determination | − |
| Gram staining | G⁻ |
| Aerobism | aerobic |
| Bacterial form | rod shaped |

Note: "−" indicates negative reaction; "+" indicates positive reaction.

The 16S rRNA gene sequence of the isolated strains was submitted to GenBank under accession number KY780223 for RW37. The RW37 strain yielded a 1438 bp 16S rRNA gene fragment. A phylogenetic tree was constructed on the basis of alignment results that were compared with available data in GenBank (Figure 2), which revealed a high degree of similarity (99%) with *Enterobacter soli*. Therefore, the RW37 strain was identified as *E. soli*.

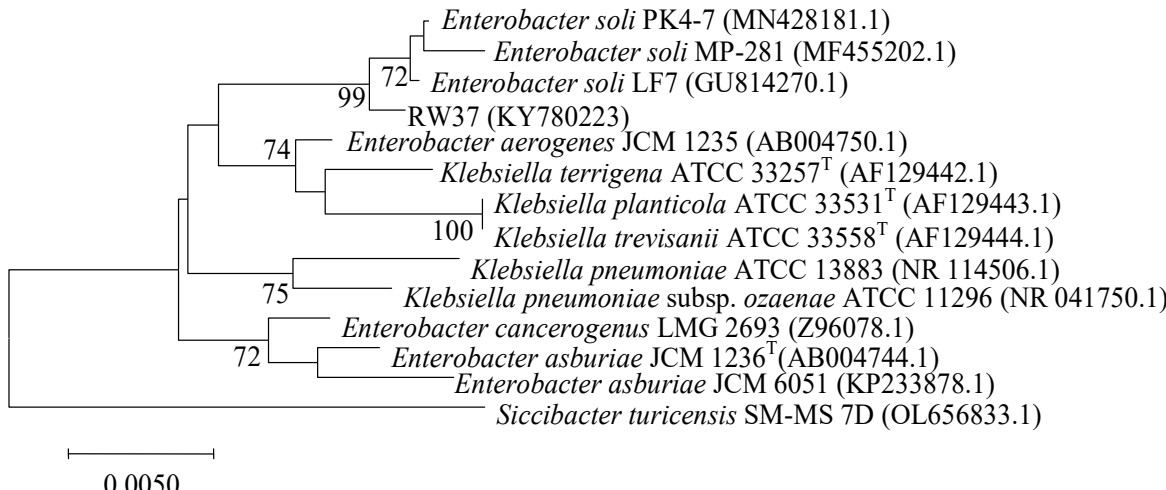

**Figure 2.** The phylogentic tree of RW37. The NJ phylogram was inferred from partial 16S rDNA sequence data. Bootstrap percentages of >70% derived from 1000 replicates are indicated at the nodes. Bar = 0.005 substitutions per nucleotide position.

### 3.2. Effects of Environmental Factors on Phosphate-Solubilizing Activity of RW37

This study found that the phosphate-solubilizing activity of the RW37 strain was significantly influenced by temperature, volume of medium, concentration of NaCl, and initial pH. The maximum activity was observed at 30 °C, and further increases in temperature led to decreases in activity. The concentration of released soluble phosphate was consistently above 278 mg L$^{-1}$ between 20 °C and 35 °C (Figure 3A). Moreover, increasing the volume of medium from 1/5 to 3/5 of the flask volume resulted in the concentration of released soluble phosphate remaining above 445.78 mg L$^{-1}$ (Figure 3B). The study found that for NaCl concentrations ranging from 1.0% to 3.0% (*w/v*), the concentration of released soluble phosphate exceeded 289.9 mg L$^{-1}$ (Figure 3C).

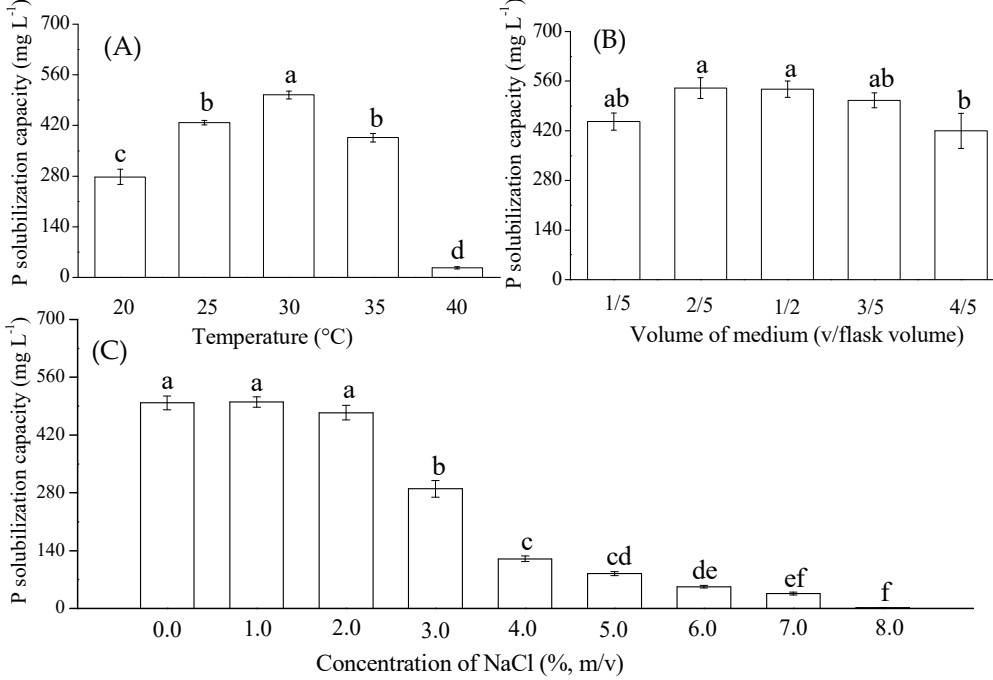

**Figure 3.** Effects of environmental factors on phosphate solubilization by the RW37 strain. (**A**): Temperature; (**B**): Volume of medium; and (**C**): Concentration of NaCl. Note: Different letters indicated significant differences among treatments at 0.05 level.

The P-solubilizing capacity of the RW37 strain varied depending on the P source and the initial pH environment (Table 3). The strongest P-solubilizing effect of RW37 was observed with $CaHPO_4$, followed by $Ca_3(PO_4)_2$, $FePO_4$, $AlPO_4$, and $C_6H_6Ca_6O_{24}P_6$. The study also found a significant interaction between P sources and initial pH environment (Figure 4). The P-solubilizing capacity of RW37 varied by up to 7-fold among the five different P sources. The pH of the fermentation broth showed significant changes under different pH conditions, ranging from an initial pH of 2.5–7.2 to a final pH of 2.34–4.45. The study found a negative correlation between the pH of the fermentation broth and the soluble P content in the Ca-P, Fe-P, Al-P, and $C_6H_6Ca_6O_{24}P_6$ treatments (Figure 5).

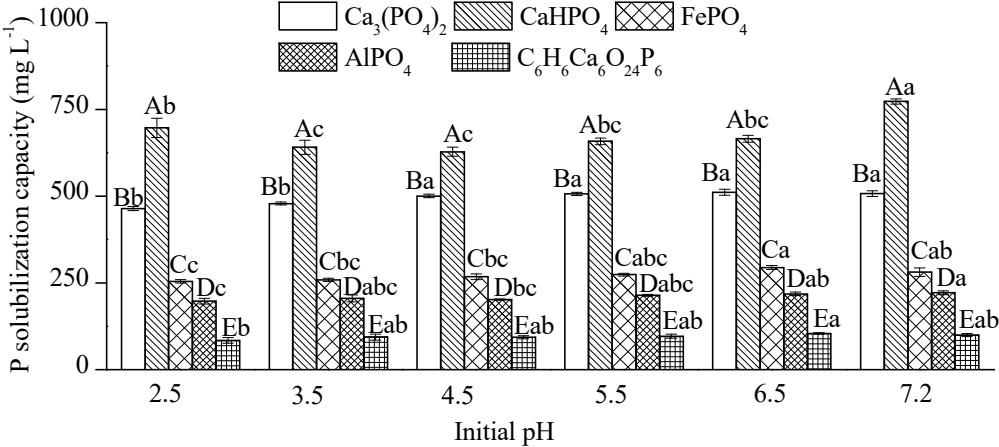

**Figure 4.** The solubilizing capacity of the RW37 strain with five P sources under different initial pHs. Note: The uppercase indicated the differences ($p < 0.05$) among the five P sources within the same initial pH and the lowercase indicated the differences ($p < 0.05$) among the initial pH within the same P sources.

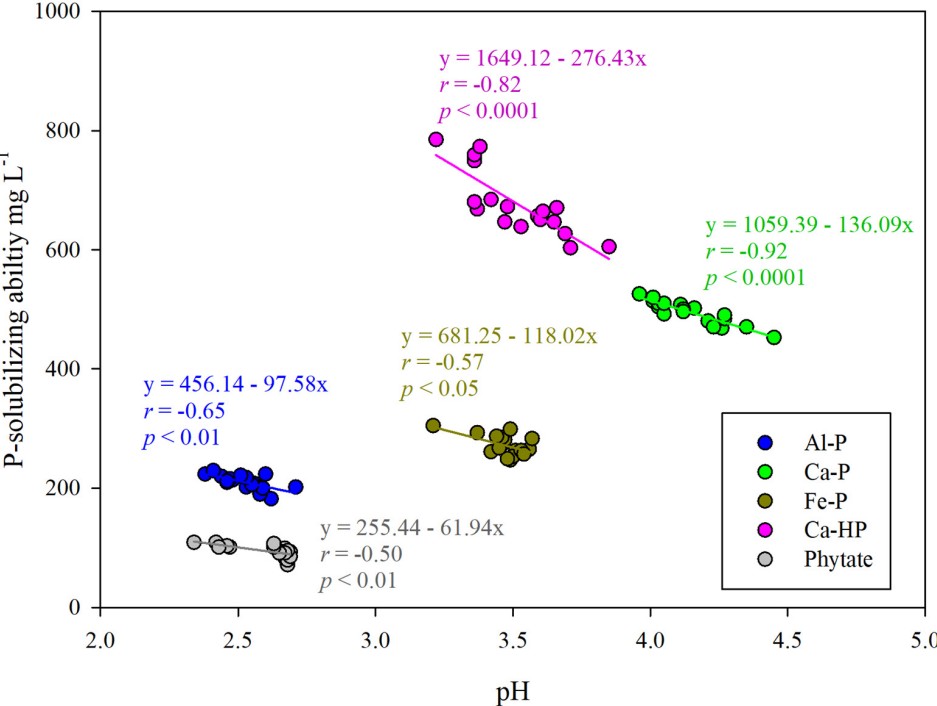

**Figure 5.** The Pearson correlations between the pH of the fermentation broth and P-solubilizing capacity of RW37 with five different insoluble P sources (Al-P: $AlPO_4$; Ca-P: $Ca_3(PO_4)_2$; Fe-P: $FePO_4$; Ca-HP: $CaHPO_4$).

**Table 3.** The *F*-values of ANOVA for the effects of initial pH and P sources on the amounts of solubilized P and pH after incubation of RW37.

| Factors | RW37 | |
|---|---|---|
| | Soluble P | Liquid pH |
| Initial pH | 14.74 *** | 4.15 ** |
| P sources | 3900.47 *** | 2296.88 *** |
| Initial pH * P sources | 6.64 *** | 5.28 *** |

Note: PSC = Phosphate-solubilizing capacity; * $p < 0.05$, ** $p < 0.01$ and *** $p < 0.001$.

*3.3. Characterization of IAA Production*

The study found that *E. soli* (RW37) has the ability to produce IAA, and its production was directly proportional to the concentration of tryptophan during different incubation periods. No IAA was produced in the absence of tryptophan. The production of IAA was highest at a tryptophan concentration of 5 mg mL$^{-1}$ after 12 days of incubation, compared to 2 mg mL$^{-1}$ and 6 days of incubation. The maximum production of IAA by RW37 was measured at 38.64 µg mL$^{-1}$ after 12 days of incubation at a tryptophan concentration of 5 mg mL$^{-1}$ (Table 4).

**Table 4.** IAA production by bacterial isolate RW37 in presence of tryptophan.

| Organism | Tryptophan (mg L$^{-1}$) | IAA Production (µg mL$^{-1}$) | |
|---|---|---|---|
| | | 6 Days | 12 Days |
| RW37 | 2 | 14.87 ± 1.15 b | 26.39 ± 2.44 a |
| | 5 | 19.18 ± 1.56 b | 38.64 ± 3.19 a |

Note: The different letters indicate the differences ($p < 0.05$) in IAA production within the same concentration.

*3.4. Effect of Inoculation on Growth and Nutrient Content of Moso Bamboo Seedlings*

The results of our study indicate that the use of the RW37 strain had a significant positive impact on the growth of moso bamboo. In comparison to the uninoculated control group (Table 5), the RW37-treated group produced a height and diameter of 37.48 and 2.24 cm, respectively, which resulted in an increase of 56.49% and 42.67% above control treatments. Additionally, we observed a significant increase in plant aboveground biomass, underground biomass, and total biomass in RW37-inoculated plants when compared to uninoculated control plants. The increase rates for aboveground biomass, underground biomass, and total biomass were 70.16%, 46.4%, and 62.19%, respectively. Inoculation with the RW37 strain significantly increased the P content in plant leaves, stems, and roots by 43.78%, 75%, and 102.33%, respectively. The inoculation also had a significant effect on the soil available P, nitrate nitrogen, and mineral N contents in the pot experiments. However, there was no statistically significant difference in the content of ammonium nitrogen between the inoculated treatment and the uninoculated control.

**Table 5.** The effect of adding the RW37 strain on the growth variables of moso bamboo seedlings.

| Item | Observed Variable | Control | Adding Strain RW37 | Increase Rate (%) |
|---|---|---|---|---|
| Plant growth | Seedling height/cm | 23.95 ± 0.71 b | 37.48 ± 1.06 a | 56.49% |
| | Ground diameter/cm | 1.57 ± 0.10 b | 2.24 ± 0.08 a | 42.67% |
| | Aboveground biomass/g | 2.48 ± 0.13 b | 4.22 ± 0.16 a | 70.16% |
| | Underground biomass/g | 1.25 ± 0.06 b | 1.83 ± 0.09 a | 46.4% |
| | Total biomass/g | 3.73 ± 0.21 b | 6.05 ± 0.27 a | 62.19% |
| Plant nutrients | Leaf P/g kg$^{-1}$ | 2.01 ± 0.11 b | 2.89 ± 0.13 a | 43.78% |
| | Stem P/g kg$^{-1}$ | 0.92 ± 0.06 b | 1.61 ± 0.22 a | 75% |
| | Root P/g kg$^{-1}$ | 0.86 ± 0.06 b | 1.74 ± 0.05 a | 102.33% |

**Table 5.** *Cont.*

| Item | Observed Variable | Control | Adding Strain RW37 | Increase Rate (%) |
|---|---|---|---|---|
| Rhizosphere soil | Available P/mg kg$^{-1}$ | $2.29 \pm 0.25$ b | $6.24 \pm 0.08$ a | 172.48% |
| | NH$_4^+$-N/mg kg$^{-1}$ | $0.96 \pm 0.09$ a | $0.81 \pm 0.10$ a | - |
| | NO$_3^-$-N/mg kg$^{-1}$ | $2.11 \pm 0.02$ b | $4.11 \pm 0.05$ a | 94.79% |
| | Mineral N/mg kg$^{-1}$ | $3.20 \pm 0.06$ b | $4.93 \pm 0.13$ a | 54.06% |

Note: Different letters indicate significant differences among treatments at the 0.05 level.

## 4. Discussion

In this study, RW37 isolated from the rhizosphere soils of *P. edulis* in acidic and P-deficient regions was identified as *Enterobacter soli.* The ability to solubilize insoluble phosphate has been confirmed in several bacteria, including *Burkholderia*, *Bacillus*, *Pseudomonas*, *Enterobacter*, and *Erwinia* [29–31]. The *Enterobacter* genus is commonly found in different soil types and may have a significant impact on the P cycle in natural environments [32,33]. In our study, RW37 was identified as *Enterobacter soli* through its morphology and multi-locus phylogeny. Surprisingly, this is the first report of an *E. soli* strain showing the potential to promote plant growth as a phosphate-solubilizing bacterium (PSB). Our results indicated that RW37 had a significant potential for converting various forms of insoluble P into soluble P. This suggests that RW37 has a strong capacity to solubilize P from inorganic pools. Notable, this capacity surpasses that seen in previous studies, which demonstrated that the *Bacillus coagulans* C45 had a low solubilizing activity of Al-P and Fe-P (9.5 and 6.4 mg L$^{-1}$) [34]. Previous studies have also shown that PSB can solubilize insoluble P, indicating their potential to promote plant growth in P-deficient soil [35]. Although previous studies have reported that some PSB, such as *Serratia* sp. S119, exhibit a strong in vitro ability to solubilize inorganic and organic P, their role in promoting plant growth is still unknown [36]. The current study demonstrates that *E. soli* (RW37) exhibits a capacity to solubilize phosphates in five insoluble forms under acidic conditions.

PSB have been reported to have the ability to utilize various insoluble P sources, including Ca-P, Al-P, and Fe-P and to convert them into soluble P forms. Ca-P is commonly found in calcareous soils, while acidic soils have a large proportion of insoluble Fe-P and Al-P. It had been observed that some PSB exhibit a high capacity for P solubilization with Ca-P, whereas others exhibit a low capacity for P solubilization with Fe-P and Al-P. Here, apart from Ca-P, RW37 notably possessed a high Fe-P- and Al-P-solubilizing capacity, revealing that RW37 could adopt a common strategy of secreting organic acids to dissolve Ca-P and could have other strategies for Fe-P and Al-P solubilization. Interestingly, the P-solubilizing ability of RW37 and the pH of the fermentation broth varied depending on the P source, indicating that the capacity and efficiency of RW37 to solubilize P is influenced by the chemical properties of the P source. Previous studies have also shown that the solubilization of P varies among different bacterial isolates and is influenced by the chemical properties of the P source [30,37]. It is noteworthy that RW37 was able to solubilize all of the tested insoluble P, suggesting the need for further investigation in fermentation and soil conditions. We noticed that RW37 exhibited the highest concentration of soluble P, accompanied by the lower pH value when provided with a P source. This observation supports previous findings that P solubilization is directly linked to a decrease in pH [3,21]. Previous studies have shown that the *Enterobacter* genus produces gluconic acid, malic acid, malonic acid, and formic acid during the solubilization of Ca$_3$(PO$_4$)$_2$ [36]. RW37 is able to establish acidic conditions to maintain its functions, and the organic acids produced may play a significant role in P solubilization. This suggests a relationship between acidification and PSB solubilization of P sources, which is supported by earlier reports [38,39]. Compared to the high Ca-P solubilization of RW37, this lower pH might not be the primary approach to solubilize Fe-P and Al-P. This result again indicated that the solubilization of Fe-P and Al-P by RW37 might be attributed to a special chelating strategy [30]. The mechanisms of P solubilization by PSB are complex and not fully understood. However, it is believed that

the secretion of organic acids by microorganisms is the main cause of P solubilization [40], and further studies are needed to validate this theory.

Moso bamboo is extensively found in the subtropical red soil regions of China [41]. Previous studies have shown that the soil pH in moso bamboo forests in southern China typically ranges from 3.5 to 5.5 [3]. Additionally, it has been reported that the red soil region where moso bamboo grows is generally acidic and lacks several essential nutrients, particularly P [3,17]. As a crucial factor in soil physical and chemical properties, pH plays a critical role in the growth and function of plants. Given these complex soil conditions, it is crucial for PSB to exhibit broad tolerance to various environmental factors in order to be widely used as a nutrient enhancer. In this study, RW37 exhibited surprising tolerance to extreme pHs ranging from 2.5 to 7.2 in culture medium and showed a relatively high ability to solubilize P in various environmental conditions. This result indicates that RW37 has a potential to adapt to a wide range of acidic soil conditions. However, further research is required to better understand the mechanism of P solubilization [42]. Nevertheless, these findings offer valuable insights for the future application of this strain in practical settings.

Previous studies have indicated that PSB inoculation has positive effects on the availability of P in the rhizosphere soil [36,42]. The increased levels of soluble P in red soils after inoculation with RW37 might be mainly attributed to an increased secretion of organic acids into the rhizosphere to solubilize and release the available P from insoluble P pools. Microorganisms could solubilize inorganic P compounds in the rhizosphere by releasing various organic compounds into the soil environment [30]. However, inoculation with the RW37 strain also increased the P content in plant tissues and promoted plant growth. In the case of red soil, inoculation with the RW37 strain has shown a significant effect on plant biomass, which is consistent with previous studies that have shown growth responses after inoculation with *E. cloacae* [36,43]. However, this increase in plant biomass does not always correspond to an increase in P concentration in plant tissues [44]. For example, Meyer et al. (2019) found that the application of *Pseudomonas protegens* CHA0 significantly enhanced plant growth but did not result in increased P uptake in calcareous soil [8]. In our own work, we found that *E. soli* induced increases of 102.33%, 75%, and 43.78% in the P content of roots, stems, and leaves, respectively, in moso bamboo seedlings. These findings suggest that the RW37 strain has the potential to improve P availability for moso bamboo growth in red soil. While it is widely acknowledged that PSB can enhance the availability of P for plants, the precise mechanism by which this occurs remains incompletely understood. Our study found that the inoculant RW37 was able to significantly increase both available soil P and plant P content. This suggests a potential mechanism for promoting plant growth, namely, the solubilization of precipitated phosphates for uptake by plants [45]. Furthermore, our study demonstrated that *E. soli* (RW37) is capable of producing IAA in pure culture. IAA is a plant hormone that is known to play a critical role in root initiation, cell enlargement, and cell division, which further supports the hypothesis that RW37 promotes plant growth [46]. Inoculation with RW37 may have resulted in the production of IAA in the rhizosphere of moso bamboo. This, in turn, stimulated the growth of root hairs and enhanced nutrient accessibility [27]. Previous studies have shown that the production of IAA by PSB in the crown region of plants can enhance nutrient uptake by the roots [30]. These findings suggest that RW37 has the potential to be used as a biofertilizer in soils with limited P availability.

## 5. Conclusions

In this study, the phosphate-solubilizing bacterium RW37 was isolated from the rhizosphere soil of moso bamboo in the acidic and red soil of Jiangxi Province, China. We identified RW37 as *Enterobacter soli* on the basis of its morphological and phylogenetic characteristics. RW37 exhibited a high activity with regard to mineral phosphate solubilization in an acidic liquid medium and also showed plant growth-promoting potential, such as through IAA production. Additionally, RW37 demonstrated high adaptability to changing environments. Inoculation with RW37 resulted in beneficial effects on the growth of moso

bamboo seedlings and also in an increase in P concentration values in both the plant and plant growth substrate. This study is the first to demonstrate the natural capability of RW37 to dissolve various types of P sources. The RW37 strain of *E. soli* has the potential to be used as a biofertilizer for moso bamboo plantations with acidic soils in China. Further research can be conducted to investigate the strain's positive effects in long-term field conditions, as well as in adverse and changing environmental conditions.

**Author Contributions:** Data curation, conceptualization, writing—original draft, and writing—review and editing Y.Z. and C.H.; supervision, writing—review and editing S.W.; supervision, investigation X.F. and F.S.; funding acquisition Y.Z., X.F. and F.S. All authors have read and agreed to the published version of the manuscript.

**Funding:** This study was financed by the National Natural Science Foundation of China (32160357 & 42067049); Jiangxi Provincial Department of Science and Technology Project (20232BBF60022 & 20232BAB205046); and the Jiangxi "Double Thousand Plan" (jxsq2023102214 & jxsq2019201080 & njxsq2020101080).

**Data Availability Statement:** The data that support the findings of this study are available from thecorresponding author upon reasonable request.

**Conflicts of Interest:** The authors declare that they have no conflicts of interest in this work.

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
