# Peer review of "Identification and Characterization of a Phosphate-Solubilizing Bacterium and Its Growth-Promoting Effect on Moso Bamboo Seedlings"

_forests, doi:10.3390/f15020364_

Round 1

Reviewer 1 Report (Previous Reviewer 2)

Comments and Suggestions for Authors

Correct this "Phosphate solubilizing bacteria"

The abstract is not clear; the author should provide highlight of the result in short way

Keywords should be more catchy and easy to understand.

The introduction section needs to be rewritten to support the reason behind this topic selection.

Add the novelty statement in the introduction section. Since this research is scientifically fill the research gap in this field to give suggestion to researchers

The sentences need valid reference "phospate-solubilizing bacteria (PSB) 48 have gained significant attention as a crucial component of PSMs" 

Add more valid references to strengthen the following statement: "Although fertilization promotes the growth of moso bamboo, while the accompanying 66 increased costs and negetive effects of chemical fertilizaers on environment force research- 67 ers to find effective alternatives technology in an eco-friendly way."

Why did the author not perform additional plant growth-promoting traits?

what are the physicochemical properties of soil sample used bacteria isolation?

How have the authors confirmed that the bamboo seedling is promoted by test bacteria only? Regarding other potential causes of bamboo seedlings, what are they?

There are so many typographical errors in the manuscript. The authors should correct.

Cross-check the references mentioned in the list to text

Comments on the Quality of English Language

The manuscript needs substantial, major revision

Author Response

Responses to Reviewer #1

Comments and Suggestions for Authors

Correct this "Phosphate solubilizing bacteria"

Response: Thank you for this suggestion. We revised it in the title, as follows: “Identification and characterization of a phosphate solubilizing bacteria and its growth-promoting effect on moso bamboo seedlings”

The abstract is not clear; the author should provide highlight of the result in short way

Response: Thank you for this suggestion. To allow readers to better understand the main conclusions and significance of this study, we have rewritten the abstract of this article. The specific modifications are as follows:

Phosphate solubilizing bacteria (PSB) offer an eco-friendly approach to boost plants growth in soils low in phosphorus (P) or deficient in it. This study isolated 97 PSB strains from the soil around moso bamboo roots in Jiangxi Province, China. The RW37 strain was identified as Enterobacter soli through its physical characteristics and genetic sequencing. Our experiments revealed that RW37 could dissolve phosphate at levels exceeding 400 mg L-1 across a wide range of environmental conditions, including temperature (25-35 oC), pH levels (3.5-7.2), salinities (0-2.0%), and volumes of medium (1/5-3/5 of flask volume), showcasing its adaptability. Furthermore, RW37 showed remarkable phosphate-solubilizing abilities in various pH treatments using different phosphate sources, with the highest capacity observed in a medium containing CaHPO4. The study also found a netative correlation between P-solubilizing capacity and fermentation broth pH, indicating that RW37 likely secrets organic acids to dissolve phosphate sources. Pot experiments demonstrated that applying RW37 significantly improved plant height, biomass, root growth, and P uptake in moso bamboo seedlings growth in red soil. Our results highlight the potential of RW37 as an eco-friendly biofertilizer for subtropical bamboo forests.

Keywords should be more catchy and easy to understand.

Response: Thank you for this suggestion. We rewrite the Keywords. As: follows:

Enterobacter soli; Phyllostachys edulis; P-solubilization ability; Growth-promoting effect; Microbial fertilizer

The introduction section needs to be rewritten to support the reason behind this topic selection.

Response: Thank you for this suggestion. Phosphorus is a vital nutrient essential for plant growth, development, and reproduction. Despite its abundance in soil, challenges arise due to the low solubility of mineral phosphorus and its fixation as iron complexes, hindering efficient phosphorus uptake by plants. More than 95% of soil phosphorus in China is insoluble. Chemical phosphorus fertilizers have been extensively utilized in agricultural and forestry ecosystems for nearly two decades to address this issue. However, the excessive application of phosphorus fertilizers escalates production costs and adversely affects soil quality. Consequently, employing environmentally friendly strategies to enhance phosphorus absorption and utilization in crops holds significant ecological and economic importance. We have rehearsed the first paragraph of the introduction section to highlight why we chose this topic.

Add the novelty statement in the introduction section. Since this research is scientifically fill the research gap in this field to give suggestion to researchers

Response: Thank you for your suggestion. Previous studies have shown that root-associated microorganisms positively impact the utilization of plant soil nutrients. Microorganisms can enhance the effectiveness of phosphorus in soil, so considering the influence of microorganisms on phosphorus cycling in soil is crucial. For example, phosphate-solubilizing microorganisms (PSM) promote plant growth by providing soluble phosphorus. Among them, phosphate-solubilizing bacteria (PSB) have received widespread attention as an essential component of PSM. Some inorganic phosphate-solubilizing bacteria can effectively convert fixed phosphorus into a biologically available form and prevent the re-fixation of released phosphorus. Previous studies have identified bacterial strains from the genera Pseudomonas, Bacillus, Rhizobium, and Enterobacter as potent phosphorus mobilizers in soil, positively affecting plant growth. However, PSB strains isolated from different soils may exhibit functional differences in their phosphorus solubilization abilities. Therefore, isolating and evaluating the environmental adaptability of PSB can provide valuable information for the practical application of PSB strains. We have rehearsed the introduction section to highlight the novelty statement of this study.

The sentences need valid reference "phospate-solubilizing bacteria (PSB) 48 have gained significant attention as a crucial component of PSMs"

Response: Thank you for this suggestion. In the revised manuscript, we added a reference to support this view. The added reference was as follows:

[8]. Meyer, G., M. Maurhofer, E. Frossard, H. A. Gamper, P. Mäder, É. Mészáros, L. Schönholzer-Mauclaire, S. Symanczik and A. Oberson. Pseudomonas protegens CHA0 does not increase phosphorus uptake from 33P labeled synthetic hydroxyapatite by wheat grown on calcareous soil. Soil Biology and Biochemistry, 2019, 131: 217-228.

Add more valid references to strengthen the following statement: "Although fertilization promotes the growth of moso bamboo, while the accompanying 66 increased costs and negetive effects of chemical fertilizaers on environment force research- 67 ers to find effective alternatives technology in an eco-friendly way."

Response: Thank you for this suggestion. We offer two references to strengthen this sentence. The references as follow:

[11]. Shahid, M., S. Hameed, A. Imran, S. Ali and J. van Elsas. Root colonization and growth promotion of sunflower (Helianthus annuus L.) by phosphate solubilizing Enterobacter sp. Fs-11. World Journal of Microbiology and Biotechnology. 2012, 28(8): 2749-2758.

[17]. Guo XM, Niu DK, Chen F, Zhang WY. Balanced fertilization and nutrient management for bamboo forest. Beijing: Science press; 2013; 27-36.

Why did the author not perform additional plant growth-promoting traits?

Response: The reviewers raised an excellent question. In this study, we primarily assessed the growth, nutrient status, and rhizospheric soil indicators of moso bamboo, including seedling height and diameter at ground level, aboveground biomass, belowground biomass, total biomass, leaf phosphorus, stem phosphorus, root phosphorus content, and soil available phosphorus, NH4+-N, NO3--N, and mineral nitrogen. These indicators also reflect the promoting effect of phosphate-solubilizing bacteria on moso bamboo growth. However, due to constraints such as the experimental period, we are conducting a series of pot- and field-controlled experiments to investigate the long-term effects of phosphate-solubilizing bacteria on moso bamboo growth in the subtropical region of China.

what are the physicochemical properties of soil sample used bacteria isolation?

Response: Thank you for this suggestion. The physicochemical properties of soil sample used bacteria isolation were listed in the Table 1, including elevation, pH, total N and Total P.

How have the authors confirmed that the bamboo seedling is promoted by test bacteria only? Regarding other potential causes of bamboo seedlings, what are they?

Response: To verify the impact of phosphate-solubilizing bacteria (PSB) on moso bamboo growth, we conducted pot experiments in this study with two experimental treatments: one without PSB inoculation and the other with PSB inoculation. Using a single-factor experiment, we sterilized the soil before the experiment to ensure the absence of viable microorganisms. By comparing the results of the two treatments, we confirmed that bacteria promote moso bamboo growth. As for other factors affecting bamboo seedlings, we plan to investigate this issue through long-term pot and field-controlled experiments.

There are so many typographical errors in the manuscript. The authors should correct.

Response: Thank you for this suggestion. We have carefully checked the manuscript and corrected grammar and spelling errors accordingly.

Cross-check the references mentioned in the list to text

Response: Thank you for this suggestion. We checked them carefully again.

Reviewer 2 Report (Previous Reviewer 3)

Comments and Suggestions for Authors

The authors attempted to revise the previous material, but some comments still remain.

 114: "The amplification protocol included one cycle of 5 min at 94 ËšC, followed by 30 cycles of 30 s at 94 ËšC, 30 s at 56ËšC, and 1 min at 72ËšC, and then one cycle of 5 min at 72ËšC" - What instrument was used for this (model and manufacturer)?

143: "The fermentation broth was centrifuged at 10,000 rpm for 10 min" - What instrument was used for this (model and manufacturer)?

146: "IAA was measured by a colorimetric technique at 533 nm" - What instrument was used for this (model and manufacturer)?

147: Where did you obtain the pure IAA? Who is the manufacturer? What is its purity?

 - How large is the solubilization zone (Figure 1A) compared to your other isolates and known strains from the literature?

 - By following the link to the 16S sequence in GenBank, we see that it was deposited there in 2017 (and for another strain, jx-09, which was isolated from soil in 2015).

 Strain jx-09 and strain RW37 the same thing?

- the presented tree and strain identification are not relevant at the moment. It is necessary to update the tree by adding typical strains.

 - When using the Salkowski method, it is not possible to specifically determine IAA, as it detects total indole compounds.

 - Is plant growth stimulation due to phosphorus mobilization or the presence of IAA?

 - There is no control group and no comparative data.

The question of the appropriateness of the journal choice still remains. With appropriate formatting, this article would be suitable for a journal like "Microorganisms" or a journal related to bacteria and their applications. There is little information about forests.

Author Response

Responses to Reviewer #2

Comments and Suggestions for Authors

The authors attempted to revise the previous material, but some comments still remain.

Response: Thank you very much for your suggestions. When submitting the manuscript, we corrected the errors in the timing of soil sampling and phosphate-solubilizing bacteria isolation. Additionally, we have modified the methods section in the revised manuscript to clarify this issue. Furthermore, we have also made revisions to the discussion section to delve deeper into the promotion effect of phosphate-solubilizing bacteria revealed in this study on the growth of moso bamboo forests in subtropical China.

114: "The amplification protocol included one cycle of 5 min at 94 ËšC, followed by 30 cycles of 30 s at 94 ËšC, 30 s at 56ËšC, and 1 min at 72ËšC, and then one cycle of 5 min at 72ËšC" - What instrument was used for this (model and manufacturer)?

Response: Thank you for this suggestion. The instrument was gene amplification instrument, and the model of the instrument was PCR T100. The PCR T100 was produced by Bio-Rad Laboratories, Inc. We have rehearsed this sentence to clarity our meaning.

143: "The fermentation broth was centrifuged at 10,000 rpm for 10 min" - What instrument was used for this (model and manufacturer)?

Response: Thank you for this suggestion. The device was high-speed freezing centrifuge. The model of the device was eppendorf 5804 and produced by Eppendorf China Ltd. We clarity this issue in the section 2.4 of revised manuscript.

146: "IAA was measured by a colorimetric technique at 533 nm" - What instrument was used for this (model and manufacturer)?

Response: Thank you for this suggestion. The device was ultraviolet spectrophotometer, and the model was Ultraviolet spectrophotometer UV-5100, the device was produced by Shanghai Metash Instruments Co., Ltd.

147: Where did you obtain the pure IAA? Who is the manufacturer? What is its purity?

Response: Thank you for your suggestion. The pure IAA was purchased by the manufacturer, and the name of manufacturer was Sangon Biotech. The concentration ≥ 99%.

- How large is the solubilization zone (Figure 1A) compared to your other isolates and known strains from the literature?

Response: The diameter of solubilization zone was about 1.3 cm compared with other isolated and known strains.

- By following the link to the 16S sequence in GenBank, we see that it was deposited there in 2017 (and for another strain, jx-09, which was isolated from soil in 2015).

Strain jx-09 and strain RW37 the same thing?

Response: Yes, the strain was deposited in 2017. The strain jx-09 and strain RW37 was the same strain except for the strain number.

- the presented tree and strain identification are not relevant at the moment. It is necessary to update the tree by adding typical strains.

Response: Thank you for your suggestion. We updated the presented tree by adding typical strains. The tree as follow:

Figure 2. The Phylogentic tree of RW37. The NJ phylogram was inferred from partial 16S rDNA sequence data. Bootstrap percentages of >70% derived from 1000 replicates indicated at the nodes. Bar=0.005 substitutions per nucleotide position.

- When using the Salkowski method, it is not possible to specifically determine IAA, as it detects total indole compounds.

Response: The Salkowski method was used for color reaction, if the color changed from yellow to pink, the result indicated that IAA could be secreted by the strain RW37. In addition, we set different concentration gradients (0, 10, 20, 30, 40, 50, 60 μg/ml) and proceed colorimetric experiment. The IAA concentration of the solution to be measured was calculated according to the standard curve. In this experiment, we found the Salkowski method was used to detect the quantitative determination in lots of previous literatures, the reference as follows:

Glickmann E, Dessaux Y. A critical examination of the specificity of the salkowski reagent for indolic compounds produced by phytopathogenic bacteria[J]. Appl Environ Microbiol, 1995, 61 (2): 793-796

- Is plant growth stimulation due to phosphorus mobilization or the presence of IAA?

Response: The mechanism of plant growth was complex, such as phosphorus mobilization, siderophore, potassium dissolving, and the presence of IAA or other growth hormone could promote the plants growth. In order to explain the mechanism of plant growth, lots of researches should be studied in the future.

There is no control group and no comparative data.

Response: We had a control group, and no IAA was produced in the absence of tryptophan in the control group. We wrote the the sentence “No IAA was produced in the absence of tryptophan” in the manuscript, and because the control treatment not produced IAA, so no comparative data was listed in the Table 4.

The question of the appropriateness of the journal choice still remains. With appropriate formatting, this article would be suitable for a journal like "Microorganisms" or a journal related to bacteria and their applications. There is little information about forests.

Response: Thank you for your suggestion. In the discussion section, we clarified that while this study involved the identification and characterization of a phosphate-solubilizing bacterium, our primary focus lies in the promotive effects of this phosphate-solubilizing bacterium on the growth of bamboo seedlings in bamboo forests. The extracted phosphate-solubilizing bacteria from this study show potential for facilitating the rapid recovery of subtropical bamboo forests subjected to human activities and natural disturbances, contributing to sustainable forest management.

Round 2

Reviewer 1 Report (Previous Reviewer 2)

Comments and Suggestions for Authors

manuscript quality is considerably improved

Reviewer 2 Report (Previous Reviewer 3)

Comments and Suggestions for Authors

The manuscript has been revised. The comments have been corrected.

This manuscript is a resubmission of an earlier submission. The following is a list of the peer review reports and author responses from that submission.

Round 1

Reviewer 1 Report

Comments and Suggestions for Authors

Comments on the Quality of English Language

Reviewer 2 Report

Comments and Suggestions for Authors

mention the total number of isolates enumerated from "bamboo rhizosphere soil" in the abstract section.

On what basis did the authors select this sample collection site "moso bamboo rhizosphere soil in Jiangxi Province"?

What are "varying environmental conditions"?

What are the physicochemical properties of the test soil?

What are the "negative environmental impacts" of overfertilization?

improve the novelty statement

Why did the author not analyze other essential elements in the test soil?

correct these "(NH4)2SO4, 0.1 g MgSO47H2O, 0.5 g FeSO47H2O, 0.5 g MnSO47H2O" and NH4+-N

how the author confirmed the test bacterial isolate was only responsible for optimistic effects on the test plant without analyzing other physicochemical properties and the microbial population in the test soil. Other factors may also play an important role in the test plant.

Why did the author not perform other PGP activities for the test isolate?

Either author performed this work in a field or greenhouse condition?

The discussion section need to be improved

Add the future perspective of this investigation

Comments on the Quality of English Language

I suggest major revision

Reviewer 3 Report

Comments and Suggestions for Authors

The topic that the authors wanted to address and partially addressed is relevant due to soil depletion of nutrients and the potential of some introduced microorganisms to promote the release of hard-to-extract components in a form accessible to plants.

Simple questions and remarks:

Line 62: Please provide more details on where exactly Moso bamboo is used and how it enhances forestry efficiency.

Line 95: "The mixture was then shaken at a speed of 120 revolutions per minute." What device was used for this (model and manufacturer)?

Line 98-99: Indexes in chemical formulas should be written in lowercase. Please mention the manufacturer of the reagents and the purity of the reagents.

Line 100: For specifying substances, it is better to choose one option. Either the formula or the verbal description (this applies to tricalcium phosphate).

Line 114: "The amplification protocol included one cycle of 5 min at 94 ËšC, followed by 30 cycles of 30 s at 94 ËšC, 30 s at 56ËšC, and 1 min at 72ËšC, and then one cycle of 5 min at 72ËšC" - What device was used for this (model and manufacturer)?

Line 143: "The fermentation broth was centrifuged at 10,000 rpm for 10 min" - What device was used for this (model and manufacturer)?

Line 146: "IAA was measured by a colorimetric technique at 533 nm" - What device was used for this (model and manufacturer)?

Line 147: Where did you obtain the pure IAA from? Who is the manufacturer? What is its purity level?

Line 161: need space between “for” and “180”

Line 174: need space between “pH meter” and “(”

Line 196: KON test or KOH test?

Line 213: adn???

Line 218: need space “to3.0%”

Line 225: Please specify substances in a consistent style. In the article, you wrote "phytate", but in Figure 4, there is a formula. Either provide the formula in parentheses.

Line 245: It should be "E. soli" in italics.

Line 298: Change "The" to "the"

Line 342: What is the word "alwway"?

- How large is the solubilization zone (Figure 1A) compared to your other isolates and known strains from the literature?

 - What is the purpose of Figure 1C (unclear and blurry), what does it mean?

 - By following the link to the 16S sequence in GenBank, we can see that it was deposited there in 2017 (and for another strain jx-09, which was isolated from soil in 2015).

 - The presented tree and strain identification are not up-to-date. When analyzing the sequence in EzBioCloud, it appears that your strain can currently be identified as Silvania confinis (99.26%).

 - When using the Salkowski method, it is not possible to specifically determine IAA, as it detects total indole compounds.

 - Is plant growth stimulation due to phosphorus mobilization or the presence of IAA?

 - There is no control group, no comparative data.

Not simple questions and remarks:

The journal chosen for presenting the results is not appropriate. With suitable formatting, this article would be more appropriate to a journal like "Microorganisms" or one related to bacteria and their applications. The article focuses more on bacteria and only briefly touches on forests.

Furthermore, a significant portion of the data has already been published online in a patent dated January 29, 2019 (which includes the figures presented in this article, Figure 1 and Figure 4, as well as Table 2). Additionally, the strain was deposited in a culture collection as 2018355 on June 11, 2018. However, in the methodology section, it is stated that the investigated strain was isolated in 2019. This leads me to suggest rejecting the publication of this article in the "Forests" journal and encouraging the authors to reconsider the results, rewrite the article, and submit it to the appropriate journal.

Reviewer 4 Report

Comments and Suggestions for Authors

Your data and your discussion do not match.

Comments on the Quality of English Language

Many typos should be corrected